# Innate Immune Pathways in Atherosclerosis—From Signaling to Long-Term Epigenetic Reprogramming

**DOI:** 10.3390/cells12192359

**Published:** 2023-09-26

**Authors:** Arailym Aronova, Federica Tosato, Nawraa Naser, Yaw Asare

**Affiliations:** Institute for Stroke and Dementia Research (ISD), University Hospital, Ludwig-Maximilian-University (LMU), 80539 Munich, Germany

**Keywords:** atherosclerosis, innate immunity, signaling pathways, trained immunity

## Abstract

Innate immune pathways play a crucial role in the development of atherosclerosis, from sensing initial danger signals to the long-term reprogramming of immune cells. Despite the success of lipid-lowering therapy, anti-hypertensive medications, and other measures in reducing complications associated with atherosclerosis, cardiovascular disease (CVD) remains the leading cause of death worldwide. Consequently, there is an urgent need to devise novel preventive and therapeutic strategies to alleviate the global burden of CVD. Extensive experimental research and epidemiological studies have demonstrated the dominant role of innate immune mechanisms in the progression of atherosclerosis. Recently, landmark trials including CANTOS, COLCOT, and LoDoCo2 have provided solid evidence demonstrating that targeting innate immune pathways can effectively reduce the risk of CVD. These groundbreaking trials mark a significant paradigm shift in the field and open new avenues for atheroprotective treatments. It is therefore crucial to comprehend the intricate interplay between innate immune pathways and atherosclerosis for the development of targeted therapeutic interventions. Additionally, unraveling the mechanisms underlying long-term reprogramming may offer novel strategies to reverse the pro-inflammatory phenotype of immune cells and restore immune homeostasis in atherosclerosis. In this review, we present an overview of the innate immune pathways implicated in atherosclerosis, with a specific focus on the signaling pathways driving chronic inflammation in atherosclerosis and the long-term reprogramming of immune cells within atherosclerotic plaque. Elucidating the molecular mechanisms governing these processes presents exciting opportunities for the development of a new class of immunotherapeutic approaches aimed at reducing inflammation and promoting plaque stability. By addressing these aspects, we can potentially revolutionize the management of atherosclerosis and its associated cardiovascular complications.

## 1. Introduction

Atherosclerosis, a chronic inflammatory disease of the arterial wall, is the primary pathology underlying cardiovascular disease (CVD), including stroke and myocardial infarction, which are major causes of disability and mortality in industrialized countries [1]. The initiation and progression of atherosclerosis involve a complex interplay of various cell types and signaling pathways, leading to plaque formation and subsequent narrowing of the arteries [2,3,4]. Activation of the innate immune pathway, mediated by concerted action of pattern recognition receptors (PRRs), is central in all stages of atherosclerosis [3]. These receptors are expressed in both immune and non-immune cells in the arterial wall, such as macrophages, dendritic cells, endothelial cells, and smooth muscle cells. They recognize specific molecular patterns associated with pathogens (PAMPs) and danger signals from damaged or infected cells (DAMPs) [5,6,7]. Among the well-known PRRs involved in atherosclerosis are Toll-like receptors (TLRs), which can recognize various ligands, including lipopolysaccharides (LPS) from Gram-negative bacteria, as well as endogenous ligands like oxidized low-density lipoprotein (oxLDL) and heat shock proteins (HSPs) [8,9,10]. TLR signaling leads to the production of pro-inflammatory cytokines, chemokines, and adhesion molecules, promoting the recruitment and activation of immune cells in the arterial wall [11,12]. A pivotal inflammatory pathway in atherosclerosis downstream of TLR signaling is the nucleotide-binding oligomerization domain-like receptor family pyrin domain-containing 3 (NLRP3) inflammasome pathway. NLRP3 is activated by various DAMPs, such as cholesterol crystals, oxLDL, and HSPs. Upon activation, NLRP3 triggers the release of pro-inflammatory cytokines, including interleukin-1β (IL-1β) and IL-18, which further contribute to atherogenesis by inducing the expression of adhesion molecules and chemokines in endothelial cells and enhancing the recruitment and activation of immune cells such as macrophages [13,14,15]. 

In addition to TLRs and NOD-like receptors (NLRs), other PRRs, including scavenger receptors and C-type lectin receptors, have been implicated in atherosclerosis [16,17,18]. These receptors recognize a variety of ligands, including modified lipids, glycated proteins, and viral RNA, and activate innate immune signaling pathways that contribute to the pathogenesis of atherosclerosis. Scavenger receptors are critical in foam cell formation and the ensuing inflammatory response. They are involved in the uptake and transport of modified lipoproteins across the atherogenic vascular endothelium and their processing by macrophages, playing a central role in both atherogenesis and atheroprogression [17]. C-type lectin receptor-mediated signaling pathways that control immune responses have been reviewed in [19]. In atherogenesis, the collective involvement of innate immune pathways is crucial, as they play a key role in promoting the recruitment and activation of immune cells in the arterial wall. Additionally, these pathways induce the production of pro-inflammatory cytokines and chemokines, highlighting their potential for therapeutic approaches to curb vascular inflammation. Hence, a deeper understanding of their precise molecular mechanisms could pave the way for innovative immunotherapeutic interventions targeting atherosclerosis. In this review, we summarize cellular immune responses in atherosclerosis and discuss innate immune pathways driving atherogenesis and atheroprogression, focusing on TLR- and NLR-dependent mechanisms and potential crosstalk with scavenger receptors. We also highlight how these pathways contribute to long-term reprogramming of the innate immune system in atherosclerosis. 

## 2. Cellular Innate Immune Responses in Atherosclerosis 

In response to inflammation, immune cells are rapidly recruited to the site of injury or infection [20]. This involves a sequential crosstalk between integrins and endothelial adhesion molecules, facilitating tethering and rolling, firm adhesion, and transendothelial migration under laminar shear stress [21]. The extent of inflammation determines which subsets of immune cells are recruited to the inflamed tissue. During acute inflammation, granulocytes like neutrophils and eosinophils are quickly recruited to neutralize and eliminate potentially harmful agents. Later, mononuclear cells, including lymphocytes and monocytes/macrophages, accumulate to aid in tissue resolution and repair [22]. However, failure to resolve the acute inflammatory condition or continuous exposure to inflammatory stimuli can lead to chronic inflammation [23]. At various stages of atherosclerosis, specific subsets of immune cells with either pro-atherogenic or atheroprotective properties accumulate. These subsets include macrophages, neutrophils, dendritic cells, natural killer T (NKT) cells, and regulatory T (Treg) cells [24]. Upon sensing danger signals like oxidized lipids, these cells infiltrate the arterial intima, initiating a cascade of events leading to plaque formation. Macrophages, for instance, engulf oxidized low-density lipoproteins (oxLDL) and transform into foam cells, a hallmark of atherosclerotic lesions. Additionally, neutrophils contribute to inflammation through the release of reactive oxygen species. Dendritic cells, on the other hand, aid in antigen presentation, further exacerbating the immune response [2]. NKT and regulatory T (Treg) cells are two specialized subsets of lymphocytes that exert contrasting effects in this context. NKT cells, primarily type I NKTs, have been implicated in promoting atherosclerosis through their pro-inflammatory cytokine release and recruitment of other immune cells to the vascular lesions [25]. Conversely, Treg cells exhibit atheroprotective properties by dampening inflammation and restraining immune responses [26]. They maintain immune tolerance and mitigate atherosclerosis progression. 

Among these cells, macrophages are the most abundant in atherosclerotic plaques, originating from circulating monocytes infiltrating the plaque or proliferation of lineage-committed resident macrophages [27,28]. These macrophages constitute a diverse and heterogeneous population, with distinct origins and varying effects on atherogenesis and atheroprogression. Advances in technology, such as genetic fate mapping, cytometry by time of flight (CyTOF), and single-cell RNA sequencing (scRNA-seq), have revealed the complexity of these macrophages and their roles in atherosclerosis [29,30,31,32,33]. Macrophages play a critical role in the initiation and progression of atherosclerosis by taking up oxLDL and releasing pro-inflammatory cytokines, which sustain the ongoing inflammation. Their activation is influenced by various stimuli, such as pathogens, tissue damage, and inflammatory cytokines, and their phenotype can adapt depending on the local microenvironment [34]. The specific contributions of immune cells to the initiation and progression of atherosclerosis have been extensively discussed in a recent review [3]. 

## 3. Heterogeneity and Function of Plaque Macrophages—Then and Now

### 3.1. The Simplified View

A previously oversimplified view was that both human and mouse atherosclerotic plaques contain two types of macrophages, M1 and M2 [35,36]. These subsets of macrophages have been extensively studied in vitro and in various mouse atherosclerosis models, leading to the notion that M1 macrophages, due to their inflammatory characteristics, promote plaque inflammation whereas M2 macrophages resolve it [37]. However, due to the varying nature of the microenvironment in the atherosclerotic plaque, such a simplified view may not exist in an atherosclerotic lesion where there are diverse and even opposing signals driving the process. Macrophages in the atherosclerotic lesion express markers associated with not only M1 and M2 phenotypes but also M4 and Mox/Mha [38,39,40]. Analysis of aortic macrophages in hyperlipidemic *Ldlr*^−/−^ mice using flow cytometry revealed distinct markers corresponding to M1, M2, and Mox phenotypes. Approximately 39% of aortic macrophages were positive for the M1 marker CD86, 21% expressed the M2 marker CD206, and 45% expressed the Mox/Mha marker heme oxygenase-1 [38]. The origin of these macrophage subsets in the plaque remains elusive. Ly6C^hi^ monocytes that infiltrate and dominate early lesions are thought to serve as precursors for M1 macrophages. However, studies using hyperlipidemic mice revealed that M2 macrophages dominate early lesions whereas M1 macrophages populate more complex lesions with an enhanced inflammatory cytokine milieu [35]. Atherosclerotic plaque milieu may be sustained by continuous influx of Ly6C^hi^ monocytes that express inflammatory cytokines. Since the vascular micromilieu can also induce a switch between macrophage phenotypes, one may envision M2 macrophages, which dominate early lesions, to switch to the M1 phenotype as the lesion progresses. Notably, M2 macrophages may be derived from the small population of lineage-committed resident intimal M2 macrophages that proliferate in the intima, rather than being recruited from the blood as reported for other inflammation-driven conditions [41]. It is important to note that M2 macrophages can exhibit distinct phenotypes and functions depending on their origin, whether they are monocyte-derived or tissue macrophages [42]. This adds to the complexity and incomplete understanding of macrophage plasticity in atherosclerotic plaques, necessitating further research to manipulate macrophages for atheroprotective functions. Altogether, the role of macrophages in atherosclerosis is far from being simplistic, and deeper investigations are required to fully comprehend and harness their potential for therapeutic purposes in atherosclerosis management. 

### 3.2. Macrophage Heterogeneity as Revealed by Emerging Techniques

The recent emergence of new technologies, including genetic lineage tracing and fate mapping, CyTOF, and scRNA-seq, has been instrumental in our understanding of macrophage origins, heterogeneity, and functions. By analyzing single cells using these techniques, distinct subsets of aortic macrophages have been identified in mice [43,44]. The main macrophage population implicated in atherosclerosis, and not found in healthy arteries, is the circulating monocyte-derived inflammatory macrophage subset, which is characterized by increased expression of inflammasome components and transcripts of cytokines and chemokines [43,45]. Pro-inflammatory type I interferon (IFN)-inducible macrophages, also derived from monocytes, express high levels of the IFN-inducible genes Ifit3, Irf7, and Isg15 [46,47]. A subset of macrophages identified during the initiation and progression of atherosclerosis, including advanced atherosclerotic plaques, are TREM2-expressing cells, which are derived from either circulating monocytes or embryonic precursors [43,44]. TREM2hi macrophages might have protective effects in vascular inflammation and resemble CD11c–YFP+ macrophages [48]. A recent study identified a subset of macrophages, known as the aortic intima-resident macrophages (MacAIR), characterized by pro-atherosclerotic effects and expression of IL-lβ mRNA [27]. These macrophages are differentiated from circulating monocytes recruited into the intima and are maintained in the aorta by local proliferation. Embryonically derived CX3CR1+ precursors are present in both atherosclerotic and healthy arteries [49]. They proliferate in the adventitia; express lymphatic vessel endothelial hyaluronan receptor 1 (LYVE1), MHC class II, and macrophage mannose receptor C type 1 (MRC1); and limit arterial stiffness by inhibiting collagen production [50]. Collectively, the utilization of single-cell analyses and lineage tracing has yielded valuable insights into the diverse characteristics of plaque macrophages, surpassing the limitations of conventional immunophenotyping. Nevertheless, achieving a comprehensive understanding of the contribution of each subset of aortic macrophages to the progression of atherosclerosis will require determining their specific roles in functional studies. Furthermore, it is imperative to thoroughly investigate their interaction with other immune cells, such as neutrophils and dendritic cells, as this will be instructive in the design of targeting strategies with minimal off-target effects. 

Regardless of their phenotypic differences, tissue macrophages function to maintain or restore homeostasis after tissue damage [51]. They navigate through a complex network of chemoattractants, including chemokines, to migrate within tissues. Cellular signaling involving PI3-kinase and the Rho family of GTPases regulates macrophage migration and chemotaxis and has been extensively reviewed [52]. The microenvironment in an atherosclerotic lesion presents challenges to both resident and emigrated macrophages. The cytokine milieu and factors like modified LDL can influence macrophage dynamics, inhibiting their migration and leading to their retention in the atherosclerotic lesion [53]. The microenvironment in the atherosclerotic lesion poses many challenges to the resident and emigrated macrophages. Besides the cytokine milieu that can skew macrophages from one state to another, there are factors like modified LDL which, when taken up by macrophages, can inhibit their migration potential, leading to their retention in the atherosclerotic lesion. Notably, neutrophils release reactive oxygen species (ROS) that contribute to lipid modification, while macrophages ingest modified lipids through scavenger receptors, transforming into foam cells as a critical event in plaque formation. Foam cells secrete inflammatory chemokines and cytokines, perpetuating inflammation. Excessive lipid uptake induces macrophage proliferation, contributing to their accumulation during advanced stages of atherosclerosis [28]. Continued scavenger receptor-mediated lipid uptake leads to the generation and intracellular accumulation of misfolded proteins, ultimately resulting in cell death. Efficient clearance of apoptotic cells (“efferocytosis”) is necessary for tissue homeostasis, and defective clearance leads to chronic inflammation. In advanced atherosclerotic lesions, cell death and defective efferocytosis contribute to the expansion of the necrotic core, potentially triggering plaque rupture and acute thrombosis [54].

## 4. Scavenger Receptors in Arterial Inflammation

### Classes, Structure, and Complexity 

Scavenger receptors (SRs) represent a highly diverse superfamily of surface-expressed proteins with little or no structural resemblance among 12 classes (class A-L) [55] (Table 1). Class A SRs contain a positively charged collagenous domain responsible for negatively charged ligand recognition and a type A cysteine-rich or a C-type lectin (CLEC) domain. Class B SRs feature a conserved CD36 domain, while class D SRs consist of mucin-like, a proline-rich center, and lysosome-associated membrane glycoprotein (LAMP) domains. Class E SRs solely contain a CLEC domain, whereas class F SRs are characterized by a high abundance of epidermal growth factor (EGF) and EGF-like domains, a short transmembrane, and an unusually long for other SRs proline- and a serine-rich cytoplasmic tail. Class G SRs are characterized by only a CXC-chemokine domain with conserved arginine residues [56]. On the other hand, class H SRs share similarities with class F receptors in terms of multiple EGF and EGF-like domains. However, class H receptors can also have fasciclin 1 and LINK domains [55,57]. 

The reason for structurally heterogeneous SR unification into a large family is due to their shared functional properties—competence in binding to mutual ligands. SRs are well known to play a crucial role in homeostasis, the most prominent of which is the recognition and removal of unwanted endogenous macromolecules, such as oxLDL and DAMPs of apoptotic cells, from systemic circulation. For example, it has been shown that SR class F, member 1 (SCARF1) can recognize dying cells through C1q-bound phosphatidylserine, which is an early “eat-me” signal on the surface of cells undergoing apoptosis and contributes to approximately 40–70% of apoptotic cell capture. Furthermore, in vivo studies demonstrated that Scarf1-deficient mice accumulated apoptotic and necrotic cells in their blood and tissue, leading to spontaneous clinical manifestation of autoimmune lupus disease [58]. In addition, members of the SR superfamily have been characterized to bind and internalize a vast range of unmodified endogenous proteins and lipoproteins, but also exogenous antigens such as PAMPs. Scavenger receptor–ligand complexes can undergo simple receptor-mediated endocytosis, trafficking through the endosome–lysosome network and resulting in either accumulation or degradation of the ligand, as well as more complex micropinocytosis and phagocytosis [59]. The diversity of endocytosis in SRs is not surprising due to sequence variations and different endocytic motifs within the cytoplasmic domains of SRs. Once internalized and delivered to endosomes, many SRs are likely to be recycled back to the plasma membrane, where they can mediate further ligand binding, clearance, or accumulation. Therefore, SRs are generally considered to be membrane-bound pattern recognition receptors, thus significantly contributing to innate immunity. 

**Table 1 cells-12-02359-t001:** The major scavenger receptors and ligands and their expression profiles.

Class	Receptor Name(s)	Ligands	Expression Profile
**A**	SR-A1/SR-A	AcLDL, oxLDL, maleylated or glycated BSA, β-amyloid, heat shock proteins and hepatitis C virus [60]; poly G and poly I, polysaccharides, including LTA and LPS, of Gram-positive and Gram-negative bacteria.	Macrophages, monocytes, mast and dendritic cells, endothelial, smooth muscle cells [60].
	SR-A6/MARCO	AcLDL, oxLDL [60], unopsonized environmental particles (TIO_2_, FE_2_O_3_, silica, and nanoparticles) [61].	Alveolar macrophages, macrophages of lymph nodes sinuses, thymus, spleen, and intestine [62], Kupffer cells in the liver [63].
**B**	SR-B1	HDL [64], LDL, oxLDL [17], apoptotic cells, hepatitis C virus [65].	Monocytes/macrophages, dendritic cells, endothelial cells, hepatocytes, and adipocytes [17,65,66].
	SR-B2/CD-36	AcLDL, oxLDL [67], HDL, LDL, VLDL, apoptotic cells [68], β-amyloid [69].	Macrophages, platelets, adipocytes, epithelial and endothelial cells [68].
**C**	SR-C1	Gram-positive and Gram-negative bacteria [70].	*Drosophila melanogaster* macrophages [70].
**D**	SR-D1/CD68	OxLDL [66], apoptotic cells.	Monocytes/macrophages [66].
**E**	SR-E1/LOX-1	OxLDL [71], C-reactive protein [72], AGE, HSP60 and HSP70 [73], apoptotic cells, activated platelets, bacteria.	Endothelial [71] and smooth muscle cells [74], macrophages, and platelets.
**F**	SR-F1/SCARF1	AcLDL [75], apoptotic cells via C1q [58], calreticulin [76], HSP70, HSP90, HSP110 [77].	Endothelial, dendritic cells, and macrophages [58].
**G**	SR-G1/CXCL16/ SR-PSOX	Phosphatidylserine, oxLDL [56], [78].	Endothelial cells, dendritic cells, macrophages [79], smooth muscle cells.
**H**	SR-H1/STAB1	AcLDL, oxLDL [80], AGEs [81], extracellular protein SPARC, TGFBi, Periostin, Reelin [82].	Monocytes/macrophages, endothelial cells [80].
	SR-H2/STAB2	AcLDL, oxLDL [80], AGEs [81], TGFBi, Periostin, Reelin [82], hyaluronan [83].	
**I**	SR-I1/CD163	Haptoglobin–hemoglobin complexes [84].	Monocyte/macrophages [84,85].
**J**	SR-J1/RAGE	HMGB_1_, β-amyloid, phosphatidylserine [86].	Endothelial cells, hepatocytes, smooth muscle cells, macrophages [86].
**K**	SR-K1/CD44	HA [87].	Monocytes/macrophages [87].
**L**	SR-L1/LRP1	VLDL [88], defensin, HSP70, HSP90.	Dendritic cells, monocytes/macrophages.

AcLDL: acetylated low-density lipoprotein; BSA: bovine serum albumin; LTA: lipoteichoic acid; MARCO: macrophage receptor with collagenous structure; HDL: high-density lipoprotein; LDL: low-density lipoprotein; VLDL: very low-density lipoprotein; LOX-1: lectin-like oxidized low-density lipoprotein receptor-1; AGE: advanced glycation end products; TGFBi: transforming growth factor, β-induced; HMGB1: high mobility group box 1 protein; HA: hyaluronan.

## 5. Scavenger Receptor-Mediated Lipid Uptake in Endothelial Cells and Macrophages

The migration of LDL through dysfunctional endothelium and the inability of macrophages to properly digest modified lipoproteins play an essential role in the development of primary atherosclerotic plaque. Until recently, passive LDL movement across a compromised endothelial barrier was thought to be the predominant mechanism for the entry of LDL into the subendothelial space and the instigation of atherosclerosis. This long-held concept was challenged by a study showing that SR-B1, a well-known receptor involved in anti-atherogenic reverse cholesterol transport by the liver, plays a crucial role in the regulation of active endothelial LDL transcytosis. It has been demonstrated that SR-B1, through its eight amino acids in the C-terminal cytoplasmic domain, physically interacts with adaptor protein DOCK4, thereby promoting SR-B1 internalization and LDL transport via RHO GTPase Rac1 activation [17] (Figure 1). 

Once in the arterial wall, macrophages recognize and engulf modified lipoproteins via several SRs, particularly SR-A1 and CD36 [89], which in turn causes the formation of lipid-laden foam cells. This perspective is based on in vitro studies depicting that SR-A and CD36 together are responsible for more than 90% of oxLDL accumulation in macrophages [90]. Studies have shown that mice with an *Sr-a1/2* deficiency fed with a high-fat diet exhibited an 80% reduction in atherosclerotic lesion area compared to the wild type [91]. Similarly, a significant reduction in lesion size has been demonstrated in *Sr-a/ApoE* double-knockout mice. Moreover, macrophages lacking *Sr-a1/2* showed a significant reduction in oxLDL uptake in vitro [59]. However, in contrast to these findings, Moore et al. demonstrated *Sr-a1*^−/−^*ApoE*^−/−^ mice fed an atherogenic diet still exhibited an accumulation of foam cells associated with an increased atherosclerotic lesion area [92]. Therefore, further research is needed to unravel the precise mechanism by which SR-A1 contributes to atherosclerosis and provide a better understanding of its therapeutic potential.

It has been reported that macrophages harvested from *Cd36*-deficient mice are defective in oxLDL uptake, and *Cd36*-deficient mice with atherosclerosis-prone background, including *LDL receptor*-null and *ApoE*-null mice, fed with a high-fat diet showed less atherosclerotic lesion formation compared to the control [18,93,94]. Another study involving bone marrow transplantation further demonstrated that the atherogenic mechanism is dependent on macrophage Cd36. Mice that received *Cd36*-deficient macrophages exhibited significantly reduced atherosclerotic lesion formation. Conversely, when macrophages expressing Cd36 were reintroduced, there was a twofold increase in the atherosclerotic lesion area. Moreover, treatment with a competitive peptide ligand (EP80317), derived from the growth hormone-releasing peptide family which specifically blocks the oxLDL binding site of Cd36, resulted in a 51% reduction in atherosclerotic lesions in *ApoE*-null mice [95]. The absence of Cd36 is not only atheroprotective by reducing lipid accumulation in macrophages, but also by diminishing the secretion of pro-inflammatory cytokines/chemokines and ROS. Additionally, *Cd36* deficiency significantly impairs the migration of macrophages in response to factors that promote lesion growth [94]. Collectively, these findings highlight the significant contribution of SRs in mediating atherosclerosis.

## 6. Downstream Signaling Events Mediated by Scavenger Receptors

SRs have been linked to a wide range of functions and are believed to play a role in complex processes such as antigen presentation, phagocytosis, and clearance of dying cells. Thus, it is not unexpected that SRs activate various signaling pathways. For example, it has been shown that SR-A1 signals through Jun N-terminal kinases (JNKs) via K63 polyubiquitylation. Specifically, triggering SR-A1 in IL-4-activated macrophages results in increased JNK activation, thereby facilitating a phenotypic switch from an anti-inflammatory to a pro-inflammatory state. Notably, this phenotypic switch was abolished upon *Sr-a1* deletion or JNK inhibition [96]. Furthermore, it has been recently established that SR-A1-mediated uptake of saturated fatty acids can also activate the JNK signaling pathway and induce IL-6 and TNFα expression [97].

Another notable linkage within the realm of SRs and signaling is evident through the interplay between oxLDL and CD36. A study illustrated that the interaction between oxLDL and CD36 mediates phosphorylation of lyn and the subsequent activation of mitogen-activated protein kinases (MAPKs), specifically JNK activation. Likewise, in hyperlipidemic mice, the absence of Cd36 resulted in reduced foam cell formation and decreased activation of JNK2. Moreover, inhibiting JNK or Src hindered oxLDL uptake and suppressed foam cell formation in vitro and in vivo [98]. Apart from mediating the uptake of oxLDL, CD36 is also involved in the regulation of cholesterol efflux through the activation of downstream Src-JNK signal transduction, resulting in the upregulation of ABCA1 expression, which in turn enhances the compensatory cholesterol efflux in macrophages [99]. Another study by Agrawal et al. revealed that CD36 regulates foam cell formation in macrophages through its interaction with STAT1 [100]. Interestingly, decreased foam cell formation in macrophages from *Stat1*-deficient mice was caused by the inhibition of CD36, but not SR-A. 

Moreover, the interaction of oxLDL with LOX-1 offers further insights into the multifaceted roles of SRs. By binding to LOX-1, oxLDL can not only be internalized but also induce the inflammatory response, intracellular oxidative stress, and cell apoptosis. In endothelial cells, oxLDL triggers cellular apoptosis by stimulating endoplasmic reticulum stress, which is abrogated by introducing an anti-LOX-1 antibody [101]. Specifically, LOX-1 induces oxLDL-mediated cell apoptosis via an elevation of intracellular NADPH oxidase (Nox-4) levels, which subsequently activates the ER stress pathway. As a result, pro-apoptotic mediators such as CHOP, Bcl-2, and caspase-12 become activated, ultimately leading to apoptosis in endothelial cells. Moreover, oxLDL/LOX-1 interaction through RhoA can enhance arginase II activation and L-arginine catabolism, leading to a subsequent reduction in nitric oxide synthase expression. This reduction in turn results in eNOS uncoupling, impaired nitric oxide synthesis, and arginase-mediated endothelial dysfunction [102]. Nevertheless, the precise mechanisms through which SRs transmit signals after binding to ligands and how these signaling pathways communicate remain to be elucidated.

## 7. TLR and NLR Signaling in Arterial Inflammation

The innate immune signaling field has been completely revolutionized by the discovery of TLRs in the mid-1990s, an extremely important finding that was awarded the 2011 Nobel Prize in Physiology or Medicine. TLRs are a family of PRRs that play a key role in the innate immune system. They recognize conserved PAMPs from foreign pathogens or DAMPs from damaged tissue [103]. Currently, 13 TLRs have been discovered in mammals, 10 in humans (TLR1-10), and 12 in mice (TLR1-9,11,12,13) [12]. All TLRs are classified as type I transmembrane receptors and share a conserved structure including an extracellular leucine-rich region that mediates the recognition of PAMPs, transmembrane domains, and cytoplasmic Toll-interleukin 1 receptor (TIR) domains essential for signal transduction [104]. The specificity of the downstream response depends not only on which TLR is activated but also on the specific PAMP, including lipids, nucleic acids, and proteins recognized by the receptors [105]. TLR4 represents a peculiar example among the group since it can recognize exogenous but also endogenous ligands with very different structures and functions, such as LPS, oxLDL, the plant diterpene paclitaxel, the fusion protein of respiratory syncytial virus (RSV), fibronectin, and many others [12,105]. Upon PAMP binding, TLRs homo- or heterodimerize, leading to the activation of a MyD88 (myeloid differentiation primary response protein 88)-dependent pathway or a TRIF (TIR-domain-containing adaptor protein-inducing IFNβ)-dependent pathway. Both MyD88 and TRIF pathways lead to the activation of the E3 ubiquitin ligase TNF receptor-associated factor 6 (TRAF6), which in turn recruits and activates the TGFβ-activated protein kinase 1 (TAK1; also known as MAP3K7) complex [106,107] (Figure 2). On one hand, active TAK1 triggers the phosphorylation of the canonical IKK complex, ultimately leading to NF-kB (nuclear factor-κB) activation; on the other hand, TAK1 is also the starting point of MAPK cascade activation. Specifically, TAK1 simultaneously phosphorylates MAPK kinases 3/6 (MKK p38 and JNKs, respectively). The third member of the MAPK family, extracellular signal-regulated protein kinases 1 and 2 (ERK1/2), is activated by IKKβ, which activates MKK1/2 upstream of ERK1/2. This cascade of events leads to the phosphorylation and subsequent nuclear translocation of several transcription factors, including cyclic AMP-responsive element-binding protein (CREB) and activator protein 1 (AP-1) that play an important role in the expression of cytokines, chemokines, and interferons. 

Among the TLRs, TLR4 stands out as the founding member of the TLR family identified as the first human homologue of the *Drosophila* Toll by Medzhitov et al. in 1997 [108]. This receptor is majorly known to be the binding partner of LPS, a membrane component of Gram-negative bacteria [104]. Its peculiarity is that it signals via both MyD88- and TRIF-dependent pathways. First, it recruits MyD88 to trigger the initial activation of NF-kB and MAPK. Subsequently, TLR4 is endocytosed and transported to the endosome, where it initiates the TRIF pathway that leads to the late activation of NF-kB and MAPK. Downstream consequences of these events include the expression of pro-inflammatory cytokines and chemokines such as IL-1, IL-6, TNFα, and IL-8 and adhesion molecules like E-selectin, VCAM-1, and ICAM-1. Due to its involvement in innate immunity and inflammation, TLR4 has been described as a crucial mediator of atherosclerosis. In 2002, Edfeldt et al. showed that the expression of TLR4, as well as TLR1 and TLR2, is markedly augmented in human atherosclerotic lesions [109]. Moreover, *Tlr4* deficiency attenuated aortic atherosclerosis, plaque lipid content, and macrophage burden when compared to *ApoE*^−/−^ controls [110]. TLR4 has been demonstrated to play a role in both the early and late stages of atherosclerosis [111]. It is well established that in the early phase of the disease, endothelial cells are activated, and this process is characterized by the expression of adhesion molecules. The upregulation of ICAM-1, VCAM, and ELAM-1, but also IL-6 and IL-8 in human coronary artery endothelial cells, depends on the LPS-TLR4 axis [112]. Activated TLR4 also plays a central role in macrophages by inducing the expression of inflammatory cytokines and proteases and it contributes to early-stage intimal foam cell accumulation at lesion-prone aortic arches in *ApoE^−/−^ TLR4^−/−^* mice compared to *ApoE^−/−^* controls [113,114]. 

Since TLRs are located at the cell surface or in endosomes, our body has developed another defense system able to recognize pathogens that have invaded the cytosol, the cytoplasmic PRRs. This family includes RIG-I-like receptors (RLRs) and NLRs [104,115]. The structure of the NLRs includes three domains: a C-terminal leucine-rich repeat (LRR) domain, an intermediate nucleotide binding and oligomerization domain (NOD, also called the NACHT domain), an N-terminal pyrin (PYD), and a caspase activation and recruitment domain (CARD) [116]. NLRs consist of 22 members in humans which are known to initiate different pro-inflammatory responses downstream of the recognition of PAMPs and DAMPs. The best-known NLRs are the ones leading to inflammasome formation. Inflammasomes are multiprotein complexes containing a sensor that recruits an adaptor molecule to activate the effector caspase-1, which catalyzes the maturation of the pro-inflammatory cytokines pro-IL-1β and pro-IL-18 [117]. Ten sensors have been discovered so far, namely NLRP1, NLRP3, NLRC4, AIM2, pyrin, NLRP2, NLRP6, NLRP7, NLRP11, and NLRP12 [14,115]. These receptors become activated by very different molecules. For example, NLRP1 recognizes anthrax lethal factor released by the Gram-positive bacterium *Bacillus anthracis;* NLRC4 is activated by NAIPs (NOD-like receptor family apoptosis inhibitory proteins), which bind to flagellin or constituents of the type III secretion system of bacteria; AIM2 detects cytosolic dsDNA; and pyrin identifies bacterial toxin-induced modifications of Rho GTPases [118]. In this regard, NLRP3 is exceptional since it is activated by a broad variety of stimuli related to bacterial, viral, and fungal infections, as well as inflammatory endogenous DAMPs [119]. Upon recognition of PAMPs or DAMPs, all inflammasomes share a common activation process where they require a sensor, the adaptor ASC (apoptosis-associated speck-like protein containing a CARD), and the effector protein caspase-1 (Figure 2). In particular, the sensor forms oligomers that recruit the common adaptor protein ASC, which in turn binds pro-caspase-1 [119]. This binding enables proximity-driven autocatalytic caspase-1 maturation and cleaved caspase-1 forms an active heterotetramer able to process pro-IL-1β and pro-IL-18. The release of mature IL-1β and IL-18 exacerbates a strong pro-inflammatory response [14,116]. Mature caspase-1 also mediates the cleavage of GSDMD (gasdermin D), a protein with a pore-forming activity that permeabilizes the membrane, leading to pyroptosis, an inflammasome-induced cell death [120]. It has also been demonstrated that GSDMD-dependent pyroptosis [121] enables the release of IL-1β and IL-18, with GSDMD pores representing channels for the secretion of these molecules [122]. 

## 8. NLRP3 Inflammasome Activation in the Pathogenesis of Atherosclerosis 

Among the inflammasomes, NLRP3 and AIM2 stand out as key drivers of atherosclerosis, with NLRP3 being the most extensively studied [13,123]. The NLRP3 inflammasome is activated via two distinct steps: priming and activation [119]. The priming phase is triggered upon recognition of PAMPs or DAMPs via receptors such as TLRs and NOD2 (nucleotide-binding oligomerization domain 2), or through the recognition of TNFα. This first step leads to the NF-kB-driven transcriptional upregulation of NLRP3 itself, pro-caspase-1, and pro-IL-1β and post-translational modification of NLRP3. The activation step requires the oligomerization of NLRP3, which together with ASC, pro-caspase 1, and NIMA-related kinase 7 (NEK7), assembles as the inflammasome. Accumulating data indicate the requirement of IKKβ for the rapid formation of the NLRP3 inflammasome and the subsequent induction of pro-inflammatory responses [124,125,126]. 

Inflammation drives atherosclerosis development and progression, and the crucial proof came from the CANTOS (Canakinumab Anti-Inflammatory Thrombosis Outcomes Study) trial, which demonstrated that targeting IL-1β with the neutralizing antibody canakinumab leads to a significant decrease in cardiovascular events compared to patients treated with a placebo [127]. Accordingly, NLRP3 involvement in atherosclerosis has been extensively investigated for its prominent role in mediating the release of both IL-1β and IL-18, which contribute to exacerbating vascular inflammation. Paramel Varghese et al. demonstrated that NLRP3 mRNA, together with caspase 1, ASC, IL-1β, and IL-18, is significantly increased in human atherosclerotic plaques compared to healthy arteries [128]. They also observed enhanced IL-1β protein release from freshly isolated human carotid plaques upon activation of TLR4 with LPS as a first signal and ATP as a second hit, providing evidence of the importance of TLR4-mediated NLRP3 inflammasome activation in atherosclerosis. Notably, the components of the NLRP3 inflammasome were found to be mainly expressed in macrophages and foam cells. This might explain why the role of the NLRP3 inflammasome in the pathogenesis of atherosclerosis is mainly studied in monocytes and macrophages. The first evidence regarding the contribution of NLRP3 in diet-induced atherosclerosis in a mouse model was from bone marrow transplantations of *Ldlr^−/−^* mice with bone marrow derived from either wild-type, *Nlrp3^−/−^, Asc^−/−^,* or *Il-1α^−/−^/Il-1β ^−/−^* mice [13]. The deficiency of these single proteins in the bone marrow significantly reduced the development of atherosclerotic lesions compared to *Ldlr^−/−^* mice transplanted with wild-type bone marrow. Recently, Christ et al. showed that *Ldlr^−/−^/Nlrp3^−/−^* double knockout mice had significantly reduced atherosclerotic plaque sizes after 8 weeks of western-type diet feeding compared to *Ldlr^−/−^* mice [123]. However, contrasting findings have been reported for the role of NLRP3 in atherosclerosis, and it is important to acknowledge the remarkable influence of factors such as gender, age, specific diet, duration of atherogenic diet feeding, and environmental conditions on the phenotype of NLRP3-deficient mice [14,15]. These should be accounted for when designing therapeutic strategies to inhibit NLRP3 for atheroprotection. 

Aside from monocytes and macrophages, accumulating data highlight the role of the NLRP3 inflammasome in other atherosclerosis-relevant cell types. Zhuang et al. showed that endothelial deletion of the transcription factor Foxp1 strongly increases the expression of all NLRP3 inflammasome components in the endothelium to promote atherosclerosis. This enhanced activation led to monocyte adhesion, migration, and infiltration, and contributed to the generation of foam cells [129]. The phenotype was reversed by pharmacological inhibition or genetic deletion of Nlrp3, demonstrating a prominent role of NLRP3 in the endothelium. Within human atherosclerotic plaques, components of the NLRP3 inflammasome are expressed in vascular smooth muscle cells (VSMCs). Active NLRP3 promotes cholesterol accumulation and foam cell formation in VSMCs through effects on HMGB1 [130]. Duewell and colleagues demonstrated that cholesterol crystals function as DAMPs in macrophages and strongly activate NLRP3 [13]. This mechanism of cholesterol-driven activation of the NLRP3 inflammasome is also operational in monocytes and neutrophils, as demonstrated by the enhanced cleavage of caspase-1 upon myeloid Abca1/g1 deficiency [121]. As interest in the application of NLRP3 inhibitors for CVD gains momentum, it becomes increasingly important to address the research gap concerning the activation of the NLRP3 inflammasome in other atherosclerosis-relevant cell types, including cell-specific blocking strategies. 

Although therapeutic blocking of IL-1 signaling has proven to be beneficial, it comes at the expense of a high rate of infections [127]. Therefore, there is the need for more specific targeting strategies, including inhibition of NLRP3. The most specific and potent NLRP3 inhibitor known to date is the diarylsulfonylurea compound MCC950, which is able to specifically inhibit NLRP3 activation in isolated mouse and human macrophages [131]. Moreover, it does not affect the function of other inflammasomes. Importantly, this compound has been demonstrated to be therapeutically effective against several preclinical models including myocardial infarction [132] and atherosclerosis [133]. MCC950 inhibits IL-1β release in bone-marrow-derived macrophages and dendritic cells stimulated with LPS and cholesterol crystals, whereas in vivo, it reduces atherosclerotic lesion development and the invasion of macrophages in the carotid artery plaques of *ApoE*^−/−^ mice [133]. Together, these data reveal the critical role of the TLR-NLRP3 axis in the context of innate immune signaling and its involvement in several inflammation-related diseases such as atherosclerosis.

## 9. Crosstalk between Scavenger Receptors and TLRs in Chronic Vascular Inflammation 

The crosstalk and synergy between different classes of innate pattern recognition receptors are essential for an effectively coordinated inflammatory response and host defense. While significant progress has been made in understanding the individual functions of these receptors, there is still limited knowledge regarding their collaborative interactions within the innate immune system. A growing body of evidence strongly suggests that SRs cooperate with TLRs as signaling partners. This cooperative network extends across various SRs, such as SR-A1, MARCO, CD36, and SCARF1, which engage with specific TLRs to modulate immune reactions. For example, it has been shown that SR-A1 and TLR4 cooperate in the phagocytosis of *Escherichia coli*, while SR-A1 and TLR2 cooperate to augment the phagocytosis of *S. aureus*. Additionally, by inducing the internalization of pathogens, SR-A1 promotes the inflammatory response mediated by endosomal TLRs like TLR3. Moreover, it has been reported that MARCO interacts with TLR2 and CD14 to recognize *Mycobacterium tuberculosis*.

Similarly, CD-36, by coupling to the TLR4/6 heterodimer, can trigger a sterile inflammatory response, as it induces expression of IL-1β and TNFα through NF-κB activation when exposed to modified LDL [134]. Moreover, it has been demonstrated that SCARF1 promotes LPS-TLR4-mediated signal transduction through NF-κB and MAP kinase pathways, leading to enhanced inflammatory cytokine release, such as IL-6, TNFα, and IFN-β [135]. In the presence of LPS, it has been observed that SCARF1 induces TLR4 translocation to lipid microdomains on the cell surface. This translocation facilitates the initiation of signaling events. Subsequently, the complex formed by LPS, TLR4, and SCARF1 is internalized into intracellular endosomes [136]. In addition, it was shown that TLR3 can interact with SCARF1 in the presence of the TLR3 ligand. Specifically, upon treatment with PIC (poly I:C, dsRNA), TLR3 and SCARF1 were found to colocalize within endosomes in THP-1 monocytes. As a result, the formation of the SCARF1–TLR3–PIC complex led to higher activation of the NF-κB pathway, as well as increased phosphorylation of MAP kinases p38 and JNK. Additionally, this complex resulted in enhanced secretion of pro-inflammatory cytokines such as IL-6 and IL-8. Thus, TLR3 acts as a coreceptor for SCARF1 and enhances its PIC-induced activation [136].

Interestingly, it has been demonstrated that SR-A and MARCO compete with cell-surface TLR4 for ligand recognition, limiting its inflammatory response, but strikingly increase responses from intracellular pathogen sensors TLR3, NALP3 (NACHT domain-, leucine-rich repeat-, and pyrin domain-containing protein 3), and NOD2. SR-A/MARCO regulates this pathway by directly interacting with the TRAF-C domain of TRAF6, thereby preventing its dimerization or ubiquitylation, which are typically required for TLR4-NF-κB-mediated activation. SR-A/MARCO-induced rapid ligand internalization prevented detection by surface TLRs whilst enhancing ligand availability in intracellular compartments, thus enabling effective sensing and robust immune responses by intracellular sensors [137]. These interactions between SRs and TLRs impact not only pathogen recognition and phagocytosis but also intracellular signal transduction and cytokine release, thus orchestrating a finely tuned immune response. As our understanding continues to evolve, unraveling the intricate interplay of these receptors promises to offer new avenues for therapeutic interventions and strategies for immune regulation.

## 10. Reprogramming of Immune Cells in Atherosclerosis 

It has been established that the innate immune system can develop a prolonged pro-inflammatory phenotype, whereby innate immune cells including monocytes and macrophages acquire memory characteristics driven by epigenetic and metabolic reprogramming. This phenomenon is termed trained immunity and it is distinct from adaptive immunity largely due to the lack of specificity, whereas it can provide protection against similar but also unrelated stimuli [138]. The growing body of evidence linking trained immunity to atherosclerosis highlights the potential of utilizing the concept of trained immunity for promising pharmacological interventions aimed at averting atheroprogression. The induction of trained immunity can be achieved through brief stimulation with β-glucan and Bacille Calmette–Guérin (BCG) vaccination for instance, or with endogenous atherogenic stimuli such as oxLDL and lipoprotein(a) [139,140,141,142]. This initial stimulation is then followed by a subsequent challenge with TLR agonists, resulting in an augmented pro-inflammatory cytokine profile, enhanced foam cell formation, increased expression of scavenger receptors, and decreased expression of cholesterol efflux transporters [143]. 

In the atherosclerotic lesion, the innate immune cells possess the capability to modify their chromatin structure by both histone methylation and acetylation. These epigenetic modifications enhance the accessibility of transcription factors regulating immune-related genes and ultimately govern the persistent pro-inflammatory phenotype [144]. At the level of histone methylation, trained human monocytes by oxLDL showed enrichment of H3K4me3 on the promoter regions of pro-atherogenic genes compared to untrained controls [141]. Additionally, isolated monocytes of patients with symptomatic atherosclerosis exhibited increased pro-inflammatory cytokine production upon ex vivo LPS stimulation compared to monocytes derived from asymptomatic atherosclerotic patients [145]. This phenotype was associated with epigenetic rewiring specifically involving histone methylation. The H3K4me3 mark is also implicated in promoting foam cell formation, which is evident through the increased expression of scavenger receptors and the decreased expression of ATP-binding cassette transporters involved in cholesterol efflux in oxLDL-trained monocytes [141]. It has been shown that training of human monocytes by oxLDL relies on histone methylation, whereby training can be completely abolished by employing methylthioadenosine, a pharmacological blocker of histone methyltransferases [141]. Another study demonstrated that statins—a lipid-lowering therapy—can prevent the in vitro training induced by either oxLDL or β-glucan, resulting in downregulation of cytokine production and epigenetic reprogramming [146]. However, a three-month statin treatment in familial hypercholesterolemia patients had no effect on epigenetic reprogramming and did not revert trained immunity [147]. 

Another significant epigenetic alteration associated with trained immunity involves histone acetylation carried out by histone acetyltransferases. The activation of innate immune cells coincides with the addition of H3K27ac to distal regulatory regions, marking active promoters and enhancers [148]. Additionally, it has been observed that the distal regulatory elements that acquire H3K27ac also typically gain the H3K4me1 mark. However, the loss of H3K27ac does not generally correlate with the loss of H3K4me1, indicating that H3K4me1 has an epigenetic memory function [148]. As in the case of histone methylation, there are significant changes in histone acetylation between healthy vessels and atherosclerotic lesions. An analysis revealed an enrichment of acetylation at two positions—H3K9 and H3K27—in macrophages, and these significant changes were increased during the progression of atherosclerosis (early versus advanced stages) [149]. Upon BCG vaccination, genes that are directly involved in inflammation and cytokine production were found to exhibit heightened levels of H3K27ac. Intriguingly, one of these genes is OLR1, which encodes the receptor of oxLDL and is considered a marker of atherosclerosis [150].

It is well established that chromatin remodeling through histone modifications plays a pivotal role in regulating the polarization of macrophages towards M1 pro-inflammatory and M2 anti-inflammatory sub-types [151,152]. Studies have been performed to examine the impact of histone modifications on macrophage polarization to understand their role in controlling the expression of inflammatory genes. For example, the levels of the H3K4me3 mark were elevated in M1 macrophages, particularly in the promoter regions of pro-inflammatory genes, when compared to M2 and naive macrophages [153]. Another example is the enrichment of the H3K4me1 mark on the IL-1β promoter, which leads to enhanced production of IL-1β, driving atherogenesis [154]. Conversely, H3K4me1 and H3K27ac were found to be enhanced by mediator 1 on M2 marker genes, exerting suppressive effects on the development of atherosclerosis [155,156]. However, in the context of trained immunity, an analysis of M1 pro-inflammatory and M2 anti-inflammatory markers in oxLDL-induced trained macrophages displayed no significant differences [141]. Similarly, with β-glucan-trained monocytes, the activation markers for both M1 and M2 macrophages were elevated without a significant difference [139]. This demonstrates that β-glucan did not skew macrophages towards a particular subtype, but rather likely triggered a global pan cell activation.

The effects of long-lasting epigenetic reprogramming were reinforced by the recent groundbreaking study highlighting the necessity of the TLR-NLR axis for immune reprogramming under atherogenic conditions. Specifically, when *Ldlr*^−/−^ mice were exposed to a western diet, broad systemic inflammation emerged; nonetheless, this inflammation became undetectable in the serum shortly after switching to a chow diet. On the transcriptional level, alterations in gene expression and epigenetic modifications within myeloid progenitor cells induced by the western diet led to an exacerbation in their rate of proliferation and an enhancement of their innate immune reactions. This demonstrates that although a western diet provoked transient systemic inflammation, it resulted in enduring modifications to myeloid cell responses. The observed effects were found to be dependent on the NLRP3/IL-1β pathway, whereby the inflammasome-dependent cytokines IL-1β and IL-18 were largely blunted in the serum of western-diet-fed *Nlrp3*^−/−^/*Ldlr*^−/−^ mice compared to *Ldlr*^−/−^ mice after 6 hours of LPS injection [123]. Collectively, these findings suggest a vital role of the NLRP3 inflammasome in the reprogramming of immune cells, particularly with respect to western diet feeding. Nonetheless, trained immunity and reprogramming in atherosclerosis is an emerging field of research [157]; thus, it is important to acknowledge that studies on atherosclerosis and related epigenetic modifications within clinical cases are limited [145,147]. Such future studies would assist in evaluating the transferability of insights gained from murine and in vitro models to clinical application in patients. 

## 11. Concluding Remarks

Dissecting the innate immune pathways in atherosclerosis has unveiled a fascinating landscape of signaling events and their profound implications on long-term epigenetic reprogramming. Through an array of intricate molecular cascades, the innate immune system controls the progression of atherosclerosis, and the culmination of these complex interactions orchestrates a delicate balance between protective inflammation and detrimental tissue damage. An investigation of key signaling molecules, such as TLRs, NLRs, and SRs, has shed light on the pivotal role of the innate immune system in initiating and perpetuating the inflammatory milieu within atherosclerotic plaques. Notably, these pathways can be manipulated for potential therapeutic interventions, aiming to either attenuate excessive inflammation or enhance protective mechanisms. Furthermore, the emerging field of epigenetic reprogramming has provided novel insights into the long-term consequences of innate immune activation in atherosclerosis. Epigenetic modifications, ranging from DNA methylation to histone modifications, contribute to stable alterations in gene expression patterns that define disease progression and complications [158]. Understanding the dynamic interplay between innate immune signaling and epigenetic remodeling offers promising avenues for developing targeted therapies that could potentially reverse or mitigate the atherosclerotic burden. However, it is crucial to acknowledge the complexities and challenges that lie ahead. Unraveling the intricacies of innate immune pathways and their epigenetic impact demands interdisciplinary collaboration, advanced technologies, and a comprehensive understanding of the complex crosstalk between different cellular components. Moreover, the translation of these findings into clinical applications requires rigorous testing in preclinical and clinical settings to ensure safety and efficacy.

In essence, the exploration of innate immune pathways in atherosclerosis, from signaling events to long-term epigenetic reprogramming, underscores the multifaceted nature of this disease and opens new horizons for therapeutic intervention. As we continue to delve deeper into the molecular mechanisms underpinning atherosclerosis, we are presented with an opportunity to redefine our strategies for disease management, ultimately paving the way for more targeted and personalized approaches that could revolutionize the landscape of cardiovascular health. 

## Figures and Tables

**Figure 1 cells-12-02359-f001:**
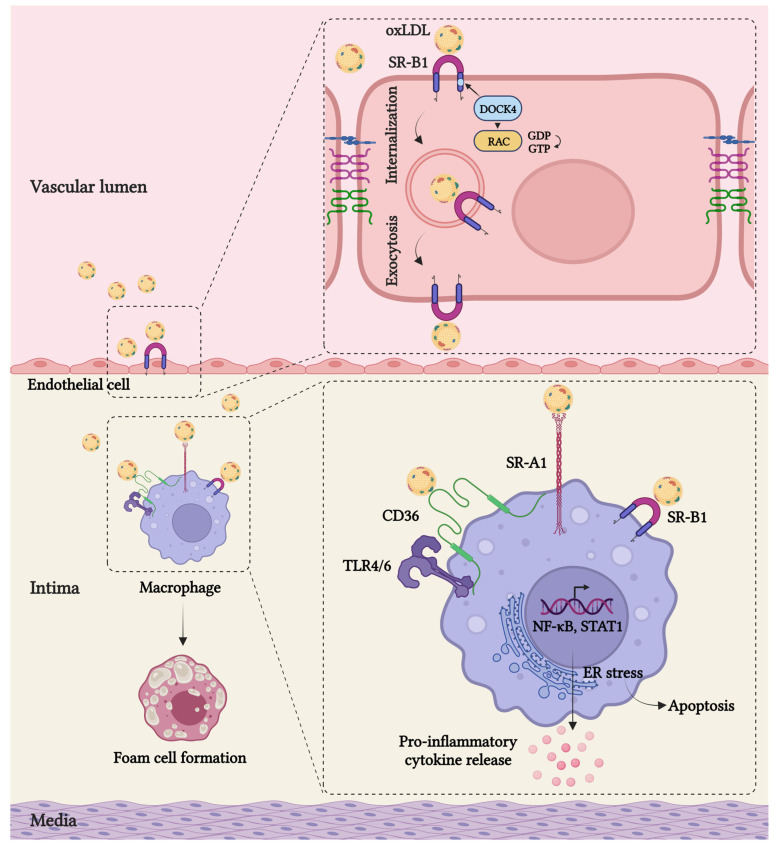
Scavenger receptors in arterial inflammation. SR-B1 mediates the transcytosis of LDL across the dysfunctional endothelium. Through its eight amino acids in the C-terminal cytoplasmic domain, SR-B1 physically interacts with adaptor protein DOCK4, thereby promoting SR-B1 internalization and LDL transport via RHO GTPase Rac1 activation. Once in the intima, macrophages play a central role by internalizing transcytosed modified LDL via SRs such as SRA1, SRB1, and CD36, which in cooperation with TLR4–TLR6 leads to the excess accumulation of intracellular cholesterol and the formation of lipid-laden foam cells, as well as the secretion of pro-inflammatory cytokines and chemokines, thereby further promoting atherosclerosis. The figure was generated using BioRender.

**Figure 2 cells-12-02359-f002:**
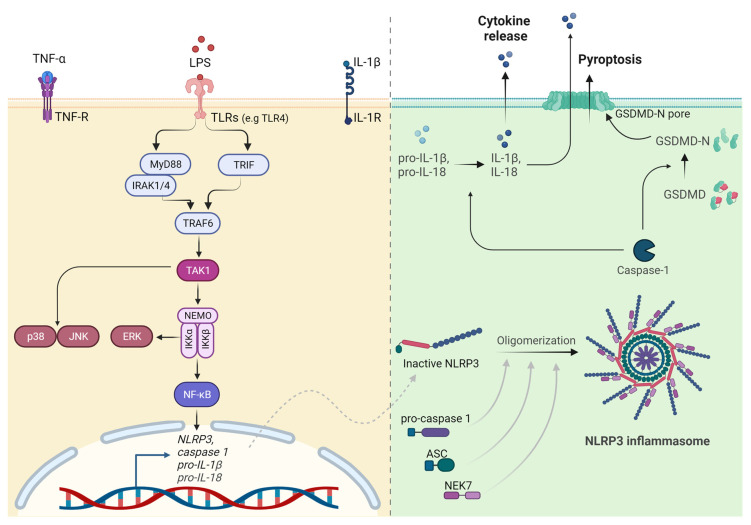
TLR and NLRP3 inflammasome signaling. Upon recognition of the respective ligands (e.g., LPS for TLR4), TLR receptors dimerize, leading to the activation of the MyD88- or TRIF-dependent pathway. In turn, MyD88 recruits IRAK1/4, and together with IRAK1 and TRIF, this leads to the activation of the ubiquitin ligase TRAF6, which has a role in the recruitment of the TAK1 kinase complex. Active TAK1 initiates the MAPK cascade, leading to the activation of p38 and JNK. Simultaneously, TAK1 also triggers the phosphorylation of the canonical IKK complex, resulting in the activation of ERK and NF-kB. On the other hand, NLRP3 inflammasome signaling can be activated by TLRs but also through the recognition of TNFα or IL-1β during the priming phase. This first step leads to the NF-kB-driven transcriptional upregulation of NLRP3, pro-caspase-1, pro-IL-1β, and pro-IL-18. The activation step then requires the oligomerization of the NLRP3 inflammasome together with pro-caspase-1, ASC, and NEK. The assembled inflammasome activates caspase-1, which in turn cleaves pro-IL-1β and pro-IL-18, releasing the mature cytokines and cleaving gasdermin D (GSDMD). This forms pores in the membrane, resulting in pyroptosis. The figure was generated using BioRender.

## Data Availability

No new data were generated or analyzed in this study.

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
