# Peer review of "Innate Immune Pathways in Atherosclerosis—From Signaling to Long-Term Epigenetic Reprogramming"

_cells, 2023, doi:10.3390/cells12192359_

Round 1

Reviewer 1 Report

The review manuscript by A. Aronova et al. describes the significance of the innate immune pathways in atherosclerosis. The manuscript is well written and focuses specifically on macrophages and three classes of immune receptors (namely scavenger receptors, TLRs and NLRs) together with their downstream signaling playing a role in arterial inflammation. The reviewed subject is clearly presented what is not easy taking into account a very wide range of scientific data available. The cited references in majority are recent and well chosen.

Two minor comments:

-          I would appreciate to read few sentences in the introduction section about other innate immune cells, such as NKT cells or Tregs, which are also known to contribute for atherosclerosis development/control.

-          The other comment regards NLRP3 inflammasome – there is a very good study form Alan Tall’s lab showing how the cholesterol efflux can impact the NLRP3 inflammasome. I would suggest to include/discuss that in the text (unless I missed it during reading).

Author Response

Reviewer #1:

The review manuscript by A. Aronova et al. describes the significance of the innate immune pathways in atherosclerosis. The manuscript is well written and focuses specifically on macrophages and three classes of immune receptors (namely scavenger receptors, TLRs and NLRs) together with their downstream signaling playing a role in arterial inflammation. The reviewed subject is clearly presented what is not easy taking into account a very wide range of scientific data available. The cited references in majority are recent and well chosen.

AUTHORS: We thank the Reviewer for the very positive assessment of our manuscript. We respond to all additional comments below point-by-point.

Two minor comments:

 I would appreciate to read few sentences in the introduction section about other innate immune cells, such as NKT cells or Tregs, which are also known to contribute for atherosclerosis development/control

AUTHORS: Thank you for this comment. Following the Reviewer’s suggestion, we now expanded on this aspect (page 2-3, paragraph 3, lines 92 – 104):” These subsets include macrophages, neutrophils, dendritic cells, natural killer T (NKT) cells, and regulatory T (Treg) cells (Engelen et al., Nat Rev Cardiol 2022, Soehnlein et al., Nat Rev Drug Discov 2021). Upon sensing danger signals like oxidized lipids, these cells infiltrate the arterial intima, initiating a cascade of events leading to plaque formation. Macrophages, for instance, engulf oxidized low-density lipoproteins (oxLDL) and transform into foam cells, a hallmark of atherosclerotic lesions. Additionally, neutrophils contribute to inflammation through the release of reactive oxygen species. Dendritic cells, on the other hand, aid in antigen presentation, further exacerbating the immune response (Soehnlein et al., 2021). NKT and regulatory T (Treg) cells are two specialized subsets of lymphocytes that exert contrasting effects in this context. NKT cells, primarily Type I NKTs, have been implicated in promoting atherosclerosis through their pro-inflammatory cytokine release and recruitment of other immune cells to the vascular lesions (Getz et al., Nat Rev Cardiol 2017). Conversely, Treg cells exhibit atheroprotective properties by dampening inflammation and restraining immune responses (Klingenberg et al., J Clin Invest. 2013). They maintain immune tolerance and mitigate atherosclerosis progression“

The other comment regards NLRP3 inflammasome – there is a very good study form Alan Tall’s lab showing how the cholesterol efflux can impact the NLRP3 inflammasome. I would suggest to include/discuss that in the text (unless I missed it during reading)

 AUTHORS: This is indeed an important aspect and thank you for drawing our attention to it. Following the Reviewer’s suggestion, we now addressed it as follows (page 15, paragraph 2, lines 503 – 507): “Duewell and colleagues demonstrated that cholesterol crystals function as DAMPs in macrophages and strongly activate NLRP3 [13]. This mechanism of cholesterol-driven activation of NLRP3 inflammasome is also operational in monocytes and neutrophils as demonstrated by enhanced cleavage of caspase-1 upon myeloid Abca1/g1 deficiency (Westerterp et al. Circulation 2018). As interest in the application of NLRP3 inhibitors for CVD gains momentum, it becomes increasingly important to address the research gap concerning the activation of the NLRP3 inflammasome in other atherosclerosis-relevant cell types including cell-specific blocking strategies”.

Reviewer 2 Report

This is an excellent review (comprehensive and well written).

Only two minor revisions are suggested:

1) It would be desirable to better identify (e.g. by numbers or other) the sub-sections related to each main topic (e.g. “Heterogeneity and function of plaque macrophages – then and now” and “Scavenger receptors in arterial inflammation”).

2) The section “Reprogramming of immune cells in atherosclerosis” should underline the current lack of studies concerning the possible relationships existing between the extension of the atherosclerotic disease (in stable clinical conditions) and epigenetic modifications and membrane phenotype of the main circulating cells of innate immunity (in particular monocytes and neutrophils).

Author Response

Reviewer #2:

This is an excellent review (comprehensive and well written).

AUTHORS: We thank the Reviewer for the very positive assessment of our manuscript. We respond to all additional comments below point-by-point.

Only two minor revisions are suggested:

1) It would be desirable to better identify (e.g. by numbers or other) the sub-sections related to each main topic (e.g. “Heterogeneity and function of plaque macrophages – then and now” and “Scavenger receptors in arterial inflammation”)

AUTHORS: Thank you very much for raising this important point. Following your suggestion, we have now higlighted these sub-sections in bold and hope this has become easy to identify.

 2) The section “Reprogramming of immune cells in atherosclerosis” should underline the current lack of studies concerning the possible relationships existing between the extension of the atherosclerotic disease (in stable clinical conditions) and epigenetic modifications and membrane phenotype of the main circulating cells of innate immunity (in particular monocytes and neutrophils).

AUTHORS: Following the Reviewer’s suggestion, we have now addressed this aspect as follows (page 19, paragraph 2, lines 647 – 657): “The observed effects were found to be dependent on the NLRP3/IL-1β pathway, whereby the inflammasome-dependent cytokines, IL-1β and IL-18, were largely blunted in the serum of Western diet-fed Nlrp3−/−/Ldlr−/− mice compared to Ldlr−/− mice after 6 hours of LPS injection [Christ et al Cell 2018]. Collectively, these findings suggest a vital role of the NLRP3 inflammasome in the reprogramming of immune cells, particularly, with respect to Western diet feeding. Nonetheless, trained immunity and reprogramming in atherosclerosis is an emerging field of research (Riksen Nat Rev Cardiol 2023), thus, it is important to acknowledge that studies on atherosclerosis and related epigenetic modifications within clinical cases are scanty [Bekkering et al ATVB 2016, Bekkering et al Cell Metab. 2019]. Such future studies would assist in evaluating the transferability of insights gained from murine and in vitro models to clinical application in patients”. 

Reviewer 3 Report

The REVIEW Innate immune pathways in atherosclerosis  from signaling to long-term epigenetic reprogrammingby Arailym Aronova et al. want to clarify the innate immune pathways in atherosclerosis, from signaling events to long-term epigenetic reprogramming, underscores the multifaceted nature of this disease and opens new horizons for therapeutic intervention. The paper is a meaningful work. However, this reviewer proposes that the manuscript could be considered the following points.

1. The authors devote a great deal of space to the activation of the inflammasome. The authors should then introduce the role of inflammasome activation in different cell lines of atherosclerosis.

2. The relationship between activation of inflammasome and reprogramming of immune cells in atherosclerosis. More discussion is necessary here.

There are some spelling and grammar mistakes throughout the paper. The English language should be thoroughly edited and polished.

Author Response

Reviewer #3:

The REVIEW “Innate immune pathways in atherosclerosis – from signaling to long-term epigenetic reprogramming” by Arailym Aronova et al. want to clarify the innate immune pathways in atherosclerosis, from signaling events to long-term epigenetic reprogramming, underscores the multifaceted nature of this disease and opens new horizons for therapeutic intervention. The paper is a meaningful work. However, this reviewer proposes that the manuscript could be considered the following points.

AUTHORS: We thank the Reviewer for the overall positive assessment of our manuscript. We respond to all comments below point-by-point.

1. The authors devote a great deal of space to the activation of the inflammasome. The authors should then introduce the role of inflammasome activation in different cell lines of atherosclerosis.

AUTHORS: Thank you for raising this important point. We would like to kindly highlight that within human atherosclerotic plaques, the components of NLRP3 inflammasome are found to be mainly expressed in macrophages and foam cells. This might explain why the role of NLRP3 inflammasome in the pathogenesis of atherosclerosis is mainly studied in monocytes and macrophages. We hope the Reviewer concur.

Nevertheless, to accommodate the Reviewer’s comment, we have now incorporated the following (page 15, paragraph 2, lines 474 – 480): “Aside from monocytes and macrophages, accumulating data highlight the role of NLRP3 inflammasome in other atherosclerosis-relevant cell types. Zhuang et al. showed that endothelial deletion of the transcription factor Foxp1 strongly increases the expression of all NLRP3 inflammasome components in the endothelium to promote atherosclerosis. In particular, this enhanced activation led to monocyte adhesion, migration, and infiltration, and contributed to the generation of foam cells [130]. The phenotype was reversed by pharmacological inhibition or genetic deletion of Nlrp3, demonstrating a prominent role of NLRP3 in the endothelium. Within human atherosclerotic plaques, components of NLRP3 inflammasome are expressed in vascular smooth muscle cells (VSMCs). Active NLRP3 promotes cholesterol accumulation and foam cell formation in VSMCs through effects on HMGB1 [131]. Duewell and colleagues demonstrated that cholesterol crystals function as DAMPs in macrophages and strongly activate NLRP3 [13]. This mechanism of cholesterol-driven activation of NLRP3 inflammasome is also operational in monocytes and neutrophils as demonstrated by enhanced cleavage of caspase-1 upon myeloid Abca1/g1 deficiency (Westerterp et al., Circulation 2018). As interest in the application of NLRP3 inhibitors for CVD gains momentum, it becomes increasingly important to address the research gap concerning the activation of the NLRP3 inflammasome in other atherosclerosis-relevant cell types including cell-specific blocking strategies”.

2. The relationship between activation of inflammasome and reprogramming of immune cells in atherosclerosis. More discussion is necessary here.

AUTHORS: Following the Reviewer’s suggestion, we have now expanded on this aspect as follows (page 15, paragraph 3, lines 623 – 633): “The observed effects were found to be dependent on the NLRP3/IL-1β pathway, whereby the inflammasome-dependent cytokines, IL-1β and IL-18, were largely blunted in the serum of Western diet-fed Nlrp3−/−/Ldlr−/− mice compared to Ldlr−/− mice after 6 hours of LPS injection [Christ et al Cell 2018]. Collectively, these findings suggest a vital role of the NLRP3 inflammasome in the reprogramming of immune cells, particularly, with respect to Western diet feeding. Nonetheless, trained immunity and reprogramming in atherosclerosis is an emerging field of research (Riksen Nat Rev Cardiol 2023), thus, it is important to acknowledge that studies on atherosclerosis and related epigenetic modifications within clinical cases are scanty [Bekkering et al ATVB 2016, Bekkering et al Cell Metab. 2019]. Such future studies would assist in evaluating the transferability of insights gained from murine and in vitro models to clinical application in patients”. 

Comments on the Quality of English Language

There are some spelling and grammar mistakes throughout the paper. The English language should be thoroughly edited and polished

 AUTHORS: Thank you for drawing our attention to this. Following the Reviewer’s comment, we have corrected all typos, edited the language, and hope it has now become acceptable.

Round 2

Reviewer 3 Report

All of the previous reviews had been solved. I therefore recommend this paper to be publicized.